# OpenReview forum: "Symbiotic Cooperation for Web Agents: Harnessing Complementary Strengths of Large and Small LLMs"
_ICLR.cc/2026/Conference — ICLR 2026 Conference Withdrawn Submission_

### Official Review · Reviewer_LDtA · 2025-10-29

**Soundness:** 1
**Presentation:** 3
**Contribution:** 1
**Rating:** 2
**Confidence:** 4

**Summary:**

This paper proposes to use the collaboration between large and small models to accomplish web agent tasks, where the large model is used to provide high-quality trajectories for distilling the small model. The distilled small model is then used for exploration, and the two models collaborate to handle tasks. They also propose a hybrid mode of large-small model collaboration to protect privacy.

**Strengths:**

I think the most reasonable part of this paper is that the hybrid mode of large and small models can handle user personalization of web services while protecting privacy. They have made some exploration in this direction, but it is very limited.

**Weaknesses:**

I do not think several claims in the paper hold, unless more evidence is provided:
* In the large-small hybrid mode, is the small model’s role really to enhance exploration? Is the exploration truly effective? As shown in Figure 3, some small models reached completely new pages, but those pages might be totally irrelevant to the task. For instance, in a “check out item” task, the model chooses to go to the “contact us” page — can this exploration really be called effective? I think stronger evidence is needed to prove that “exploration is effective,” not merely that “exploration increases.”
* The two technical changes, Speculative Data Synthesis and the previous small-model exploration claim, are contradictory (using data selected by the small model for retraining is somewhat like rejection sampling — it reinforces already learned knowledge and does not really strengthen exploration). Multi-task Learning is not novel (many current web agent works already train high-level reasoning/planning and low-level action prediction, which correspond here to rationale and action prediction).
* I’m surprised that you did not create your own training tasks and data and then test on WebArena, but instead used this benchmark for both training and testing. Based on this, your baselines should be web agent works that train and test on the same benchmark (e.g., works related to agent memory or those aiming to validate agent evolution), rather than baselines that were never trained on the test cases. The so-called state-of-the-art comparison is therefore not fair.

**Questions:**

Do you have more exploration on the large-small hybrid mode, such as for web service personalization? And can you find reasons to rebut the points mentioned in the weakness section above? I am willing to adjust my score if you can fully convince me.

---

### Official Review · Reviewer_4ere · 2025-10-30

**Soundness:** 3
**Presentation:** 2
**Contribution:** 2
**Rating:** 2
**Confidence:** 4

**Summary:**

Web Agents are typically trained by generating trajectories to explore a web environment with a large language model (LLM) which are then used to distill knowledge into a small language model (SLM). This paper proposes are new training framework in this context. The key innovations are the following:
- For the distillation process the SLM  is used for speculative generation to mitigate the off policy distillation bias.
- A multi task learning approach is employed that jointly learns actions and reasoning steps to fully leverage the capabilities of the large language model
Furthermore the authors introduce a privacy preserving mode that directs tasks that might contain privacy data to a local SLM.

**Strengths:**

- Significance of results: 48.5% SR on webarena with an 8B parameter model is a major advancement. This result effectively closes the performance gap between small, deployable agents and large, API-based agents, which is a highly significant contribution to the community.
- Novel Framework Concept: The "symbiotic loop" is a novel and intuitive concept. The idea that a SLM isn't just a "worse" version of a LLM, but a complementary partner that can contribute new exploratory data back to the LLM is interesting. This is well supported by the page visitation analysis in Figure 3 and the increasing synergy metric in Figure 6.

**Weaknesses:**

See questions

**Questions:**

Empirical Validity:
- A critical issue undermining the paper's empirical claims is a note apparently left in Appendix R (Table 4). The quoted text ("As the exact baseline number for GPT-4.1 was not specified in the rebuttal, we have used hypothetical values...") is extremely concerning. It suggests that the manuscript is an incomplete draft and, more alarmingly, that some of the reported results are "hypothetical values" rather than genuine experimental data. This fundamentally compromises the paper's scientific validity. The authors must immediately clarify this and confirm that all figures reported in the paper are from final, completed experiments.
- There seems to be an inconsistency in the baseline reporting between Table 1 and Table 2. Table 2 establishes a "vanilla supervised fine-tuning (LLAMA-8B)" baseline with a 40.8% SR, which serves as the true starting point for the paper's distillation enhancements. However, Table 1, which summarizes the main results, omits this 40.8% SFT baseline. Instead, it only lists a "Vanilla prompting" llama 8B baseline (5.6% SR). This comparison seems to be misleading. For a fair and transparent evaluation, the 40.8% SFT baseline should be included in Table 1 as the primary point of comparison for the SLM results.

Requests for Clarification:
- The description of "Speculative Data Synthesis" (SDS) in Section 3.3 and Figure 4(a) seems to constrain the SLM's exploratory role. In this phase, the SLM proposes an action that is validated against the LLMs topk candidates. If the proposal is rejected, the teacher's action is used. This appears to be more of a teacher-guided filtering step than autonomous exploration. Could the authors clarify how this SDS mechanism relates to the separate step "Small LLM exploration" where the SLM is described as uncovering "diverse trajectories" on its own?
- The authors introduce "Speculative Data Synthesis" (SDS) as a key innovation. However, the core concept of using student-generated proposals that are then filtered by a teacher model  to mitigate off-policy bias shares significant similarities with existing on-policy distillation and speculative decoding/execution methods (e.g., as seen in recent technical works like the Qwen 3 Technical Report [1]). The authors should provide a more thorough discussion of related work in this specific area to better contextualize their contribution and highlight its precise novel aspects.

Questions on Practicality and Framing:
- The practical implementation of the "Hybrid Mode for Privacy Preservation" raises efficiency concerns. The paper states that a "local DeepSeek-R1 model" (a 32B model per Section 4) is used only as a "Privacy Detector", while a separate local SLM (e.g., LLaMA-8B ) is used to handle the private tasks. This approach necessitates deploying and running two different models locally. The authors should justify this design. Why not use the 32B DeepSeek-R1 model, which is already deployed for detection, as the local agent itself? Conversely, could the smaller local agent not be fine-tuned to also perform the privacy detection task?
- The "Hybrid Mode for Privacy Preservation" feels disconnected from the paper's core 'symbiotic' contribution. The conclusion claims this mode "leverages the complementary strengths of large and small LLMs" , but the described mechanism is a simple router that delegates tasks to a local small model for privacy or a cloud-based large model for performance. This is a practical trade-off, not an example of the symbiotic improvement cycle that is central to the paper's main contributions. The framing of this feature feels like an overstatement.

---

> ### Author Response · Authors · 2025-11-19
>
> We sincerely apologize for the confusion and valid concern caused by the erroneous note found in Appendix R. We fully understand how this oversight compromises the perception of our work's scientific validity, and we are grateful to you for identifying it.
>
> **1. Explanation of the Error**
>
> The text in question ("hypothetical values") was an artifact of a clerical error during the manuscript preparation phase. We were in the process of integrating additional experimental results prepared for a previous conference rebuttal into this submission. To facilitate the formatting and merging of tables, we utilized an LLM assistant. Unfortunately, the model hallucinated the placeholder note, and we failed to catch and remove this text during our final proofreading before submission.
>
> **2. Correction with Actual Experimental Data**
>
> We wish to explicitly clarify that the experiments regarding GPT-4.1 were **fully conducted** and are **not hypothetical**. The data is genuine. The correct experimental results, which should have replaced the placeholder text in Table 4, are as follows:
>
> *   **Model:** GPT-4.1
> *   **Metric:** Success Rate (%) on 1/10 subset
> *   **LLM-L (Baseline):** 47.5%
> *   **AgentSymbiotic (Ours):** **52.5%**
>
> These results align with the performance trends observed in other models, demonstrating a consistent improvement (approx. +5%) attributed to our framework.
>
> **3. Verification**
>
> Following your comment, we have conducted a comprehensive audit of the entire manuscript and all appendices. We confirm that this was an isolated editing incident restricted to the caption of Table 4 in Appendix R. No other data points, tables, or claims in the paper are affected by this issue. We will update the manuscript immediately to reflect the correct values and remove the erroneous text.
>
> Thank you for your rigorous review, which has been essential in ensuring the accuracy and integrity of our final manuscript.

---

### Official Review · Reviewer_vS1h · 2025-10-30

**Soundness:** 2
**Presentation:** 2
**Contribution:** 2
**Rating:** 4
**Confidence:** 4

**Summary:**

This paper proposes AgentSymbiotic, a framework that creates an iterative relationship between large and small LMs for web browsing tasks. The core idea is as follows:
1. A large LM (e.g., GPT-4o) generates high-quality trajectories
2. A small LM (e.g., Llama-3 8B) is distilled from this data
3. The distilled small LM explores the web environment to discover diverse trajectories the large LM might miss
4. These trajectories are added to a RAG knowledge base that improves the large LM in the next iteration.

It introduces two additions to distillation: speculative data synthesis (SDS) to mitigate off-policy bias and multi-task learning to preserve reasoning capabilities. A hybrid privacy mode delegates sensitive tasks to local small LMs after passing them through a separate local LLM that assesses the content and potential for sensitive/private information.
The main contributions are:
- The symbiotic framework coupling large and small LLM improvement
- Novel distillation techniques (SDS + multi-task learning)
- SOTA results on WebArena for small LM in a big improvement over past SOTA

**Strengths:**

- Outstanding empirical results: Achieving 49% SR on WebArena with an 8B model (vs. 28% prior best) is a major leap, demonstrating that much smaller models can achieve near-large-model performance on complex web tasks.
- Novel framework: Using a cheap, fast small LLM as "explorer" to generate diverse data for a powerful, expensive large LLM "exploiter" is elegant and intuitive.
- Technical innovations: SDS addresses off-policy bias with theoretical justification and multi-task learning preserves reasoning capabilities during distillation. Both are well-motivated solutions to known problems.
- Practical contributions: The hybrid privacy mode (89.8% F1-score) addresses real deployment concerns. Evaluation includes domain-specific analysis and generalizability across different LLM backbones.

**Weaknesses:**

**Major Issues**
- Transductive evaluation undermines generalization claims. While the authors support this by mentioning this is the case in other papers, I think this undermines the novelty of the method. There are few high-quality web agent benchmarks and "burning" them to train seems to defeat the purpose of benchmarks in my opinion. The paper trains and tests on the same 812 WebArena tasks, with trajectories from these tasks used for both distillation and RAG. While acknowledged in Appendix S as "transductive learning," this makes it hard to distinguish genuine reasoning from memorization. The overlap analysis removing 19 "identical" trajectories seems insufficient. Trajectories can be semantically near-identical retrieval of solutions to exact test tasks resembles data leakage rather than generalization. Key question: Does the RAG database contain successful trajectories for the exact task being tested?
- Exploration claim. The core premise that small LLMs are superior explorers due to "distinct reasoning capabilities" or "increased stochasticity" appears weak to me. This "divergence" could simply be errors or noise, not novel solutions. Importantly, could this behavior not be replicated by increasing sampling temperature in the large LM? Without direct comparison to high-temperature large LLM exploration, the "symbiotic" claim is unfounded. The gains may come from "stochasticity + RAG" rather than any small LM-specific property.

**Presentation issues**
- Formalism needs a solid re-write. For example, section 3.1 uses T for both a task and the task set, refers to trajectories as both H and τ inconsistently, and presents standard concepts (Eq. 3) with unnecessarily complex explanations.
- Some important methodological details are missing. The "multi-LLM debate" and "predefined criteria" for trajectory evaluation (Section 3.2) are fundamental but are not explained in the main text/referenced as appendix sections.
- Citation style and presentation: Non-standard inline citations disrupt reading flow. Redundancy in text (first paragraph of Section 3.1 repeats exploration/exploitation points).

**Questions:**

- I think you should provide evidence of inductive generalization. What is performance when training on tasks 1-400 and testing on 401-812? Without this, claims reflect "SOTA memorization" rather than "SOTA performance."
- Provide direct experimental comparison between small LM exploration and self-iterating large LM with high sampling temperature (T=0.8-1.0). This is an important missing baseline.
- When solving task X, does the RAG database contain other successful trajectories for task X? If so, how is this different from providing solved examples of the exact test problem?
- What criteria does the multi-LLM debate use? How is the debate structured? How many LLMs participate? I feel like there should be an intuition of this in the main text.

---

### Official Review · Reviewer_oYaq · 2025-11-02

**Soundness:** 3
**Presentation:** 4
**Contribution:** 3
**Rating:** 6
**Confidence:** 3

**Summary:**

The paper proposes AgentSymbiotic, a framework where large and small LLMs improve each other through an iterative process. Large models generate high-quality trajectories for distillation, while small models explore more diverse paths that feed back into the large model’s RAG system. The method also introduces speculative data synthesis to reduce off-policy bias, multi-task distillation to keep reasoning ability, and a hybrid mode for privacy.

**Strengths:**

The idea of having large and small models cooperate and tracking their joint improvement with a “synergy” metric is interesting and well motivated. The paper also provides some theoretical grounding for Speculative Data Synthesis, showing how it can tighten the student–teacher policy gap. The hybrid mode for privacy is a practical and thoughtful addition that makes the system more realistic. Experiments are extensive, covering several models and detailed ablations while limited to only WebArena benchmark. Overall, the paper is well written and clearly presented, with nice visualizations that make the framework easy to understand.

**Weaknesses:**

**Major comments:**

- The main components (teacher–student speculative data synthesis, incorporating reasoning tokens during distillation, and RAG-enhanced large models) are not individually novel. However, integrating them in a systematic, iterative framework is a nice contribution.

- From Figure 3, it is not clear how the authors conclude that smaller models are more explorative than large models. In fact, it seems that Claude-3.5 visited more unique pages than others. So, maybe focusing on cost efficiency of small LLMs versus large LLMs could be a better motivation behind the symbiotic loop.

- Appendix Q seems quite important; it might be worth bringing part of it into the main paper. It would also help to clarify whether LLM-L + RAG is equivalent to the “symbiotic” setup but with the large model taking actions in the environment as well.

- Table 2 shows that improvements from the multi-task and speculative components are not uniform across sub-domains; in some cases, performance even drops. It would be useful to include a short discussion on why this happens.

- One key limitation of the paper is that all experiments are conducted only on WebArena. While the results are kind of strong, it remains unclear whether the observed improvements would transfer to other domains. Evaluating on at least one additional benchmark would make the claims more convincing.

**Minor comments:**

- Use \citep when the reference is not part of the sentence.

- Add a few missing references for fine-tuning small LLMs for web tasks (e.g., [1,2]).

- [Line 391] Why was a subset of tasks used? It should be straightforward to reproduce the figure on the full benchmark.

[1] Yu et al. Demystifying Reinforcement Learning in Agentic Reasoning

[2] Vattikonda et al. How to Train Your LLM Web Agent: A Statistical Diagnosis

**Questions:**

- [Line 224] How important is the multi-LLM debate? If it’s key, maybe include it in Figure 2. Did you try with a single LLM to see the difference?

- [Line 255] What are the predefined criteria for selecting trajectories?

- In Table 1, when reporting results for small models, is the large model always Claude-3.5? And when the large model is Claude-3.5, which small model is used?

- [Line 430] Again, why only show a “representative subset” of tasks? Reporting all tasks should be straightforward.

---

### Note · Authors · 2025-12-06

I have read and agree with the venue's withdrawal policy on behalf of myself and my co-authors.